# Combining Pultrusion with Carbonization: Process Analysis and Material Properties of CFRP and C/C

**Jonas H. M. Stiller** [1,*] ◉, **Kristina Roder** [1], **David Löpitz** [2] ◉, **Marcus Knobloch** [2], **Daisy Nestler** [1] ◉, **Welf-Guntram Drossel** [2] **and Lothar Kroll** [1]

[1] Institute of Lightweight Structures, Chemnitz University of Technology, 09107 Chemnitz, Germany
[2] Fraunhofer Institute for Machine Tools and Forming Technology IWU, Reichenhainer Straße 88, 09126 Chemnitz, Germany
* Correspondence: jonas.stiller@mb.tu-chemnitz.de; Tel.: +49-371-531-35354

**Abstract:** Composites made of carbon-fiber-reinforced carbon (C/C or CFC) are high-performance materials with a wide range of properties, making them especially suitable for the design of thermally and mechanically highly stressed components. As the production process of these high-performance materials is currently still very expensive, new concepts for an economical manufacturing process are required. This paper focusses on an innovative approach that uses the polymer-based pultrusion process for shaping with a subsequent carbonization step to C/C. In this process, carbon fibers (CF) and a phenolic resin were used to manufacture a semi-finished product made of unidirectional (UD) carbon-fiber-reinforced plastic (CFRP) with a fiber volume content of 66%. The C/C composite shows dimensional stability and has a flexural strength of approx. 240 MPa and a flexural modulus of approx. of 135 GPa with an elongation of 1.8%.

**Keywords:** phenolic resin; pultrusion; CFRP; C/C; carbon matrix; carbon fiber; composite; CF





## 1. Introduction

Carbon-fiber-reinforced carbon (C/C) composites consist of carbon fibers (CF) embedded in a carbon matrix and are characterized by unique properties such as low density, high thermal conductivity, high stiffness, and good fracture toughness. They can be used in a high temperature environment above 2000 °C under an inert atmosphere. However, they are susceptible to oxidation in an oxygen-containing atmosphere above 500 °C. C/C composites are used mainly for specialized military and aerospace applications, such as exit cones, thermal protection of space shuttles, and nose tips of reentry vehicles. They can also be found in industrial applications, including aircraft brakes, jet engine parts, feed pipes in the chemical industry, hot-pressing dies, parts in nuclear reactors, components, and heating elements for high-temperature furnaces; C/C composites are also found in the energy sector as polar pates in fuel cells and charging systems for the heat treatment of metal parts [1–4].

To produce C/C composites, three main production routes exist. In two processes, a CF preform is infiltrated with liquids (thermosets, thermoplastics); in the third process, the preform is infiltrated with a vapor. Thermosetting resins, such as phenolic resins, and thermoplastics, such as pitches, can be used as liquid matrix precursors to infiltrate the fiber material. This infiltration step is followed by a subsequent pyrolysis or carbonization step with temperatures around 1000 °C or higher. When thermosetting resins are used, during infiltration also the shaping of the component takes place, for which a wide range of processes are available. These shaping processes include, for example, hand lamination, the autoclave technique, the filament winding process, resin transfer molding (RTM), and the pressing technique. Depending on the thermosetting resin, one or more additional re-impregnation and re-carbonization process cycles might be necessary to achieve better

material properties of the C/C composites. Furthermore, a final heat treatment step (graphitization process at about 2400 °C) can further enhance the properties of the composite. In addition to the processes based on liquid polymer impregnation, a carbon matrix can also be infiltrated and deposited from gaseous precursors into a fiber preform via a gas phase process (chemical vapor infiltration—CVI) [2,5–8].

Despite their unique properties, the material class of C/C composites are used in few products due to their high processing costs, which make the broad application expensive and uneconomical. As a result, many components are manufactured only for prototypes and small series. Thus, reducing the processing costs is a fundamental requirement for a more economic production and widespread use of C/C composites. One approach to lower processing costs is the utilization of processes that are suitable for large series production, such as polymer-based molding processes. The advantages of those processes lie in their high level of automation and resource conservation due to near-net shaping leading to energy and cost savings.

Polymer-based large-scale manufacturing processes offer an opportunity for cost-effective shaping due to their usually large output and have already been implemented in processing fiber-reinforced ceramics. For example, injection-molding technology is used to process compounds made from carbon-fiber-reinforced phenolic resins that are used for the liquid silicon infiltration process (LSI) [9,10]. In this process, a thermoplastic processable novolak is mixed with the hardener urotropin and CF and compounded into granulate. This granulate is remolten in the injection molding cylinder and injected into the hot cavity, where the three-dimensional crosslinking reaction takes place [11]. The subsequent pyrolysis is performed in an inert atmosphere at temperatures of about 1000 °C, at which a conversion of the polymeric into a carbon matrix occurs. Due to the shear forces introduced into the material by the screws, especially in the twin-screw extruder for the compound production and in the injection molding machine, the fiber length of the initial fibers is significantly reduced [12,13]. A possible approach to increase the mechanical properties can be realized using textile inserts in the injections molding cavity. Another option is the utilization of compounds made from silicon carbide-particle- and carbon-fiber-reinforced polysilazanes [14] for the liquid polymer infiltration process (LPI). The injection molding process in particular enables a high level of geometric complexity in the parts. Furthermore, the extrusion process [15] is applied to produce profiles for the LSI process, which consists of a thermoplastic and semi-coke as binding agents and activated carbon powder and CF as fillers.

By means of the composite flow molding process [16], a pultruded rod of carbon-fiber-reinforced polyether ether ketone (PEEK) is formed into a new shape and further processed by the LSI process. However, the pyrolysis must take place in a mold because, in contrast to the thermoset phenolic resin, dimensional stability is not given when the PEEK is heated in itssoftening range.

The pultrusion process is one of the few manufacturing processes for continuous fiber-reinforced plastic profiles that is suitable for large-scale production. The minimal waste and the very high mechanical properties make pultrusion profiles a compelling option for the shaping step of the C/C composite production. Pultrusion profiles are available in a wide range of standard geometries such as flat profiles on coils (e.g., for wind power rotor blades [17]) as well as hollow complex multi-chamber profiles in the infrastructure sector [18]. However, component design is limited to constant profile cross sections only.

Figure 1 shows the basic features of the pultrusion process. Rovings or semi-finished fiber products (1) are pulled from creels by alternately moving pulling devices (4) and pass through an impregnation unit (2). In the next step, the impregnated reinforcements are pulled through a heated die (3), in which the liquid thermoset resin cures completely within a few seconds. A saw (5) cuts the profiles to the desired length [19].

Due to the process and the high degree of automation, mainly continuous fiber reinforcements in the form of rovings or textile fabrics are used in the pultrusion process. This enables a continuous operation and allows the processing of low-cost textile-based raw

materials. As far as matrix systems are concerned, mainly thermosets such as unsaturated polyesters, vinyl esters, or epoxy resins have been used. Recently, increasing demands on cost-effectiveness and performance have also led to the establishment of highly reactive polyurethanes as matrix materials [20]. The phenolic resins typical for C/C production have only a very minor role in pultrusion. Due to their exceptional fire, smoke, low smoke toxicity (FST), and heat resistance, the use of phenolic resins in combination with glass fiber reinforcement is in demand in special fields, such as for passenger transportation. Pultrusion as a starting process for C/C material production has already been considered. In [21], Link et al. produced CFRP rods using a specially developed microwave curing process. The rods are then carbonized and graphitized, and their dimensional stability is investigated. Link et al. focused on the microwave curing technology; thus, only limited information on the C/C material used in their study is available.

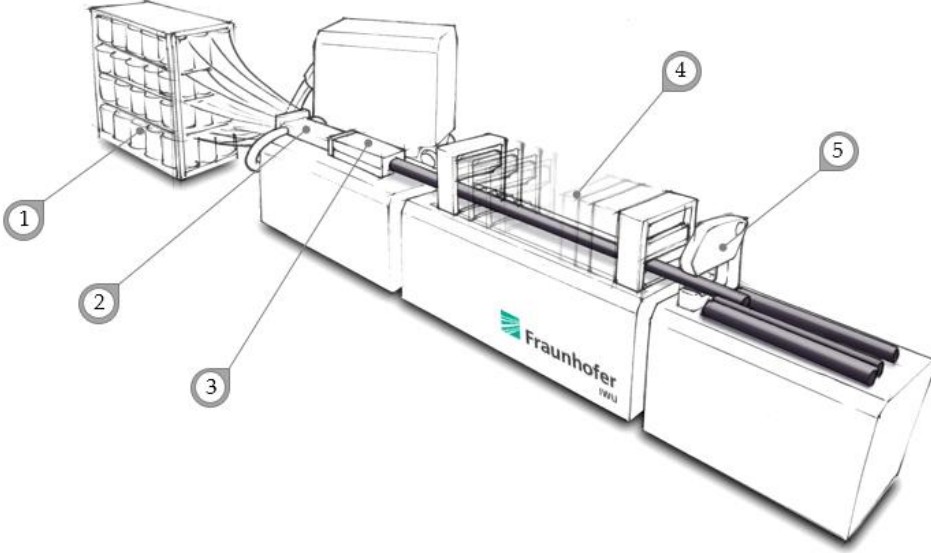

**Figure 1.** Pultrusion line: (1) bobbin creel; (2) impregnation unit; (3) heated die; (4) pulling devices; (5) saw.

Therefore, the purpose and novelty of this study was the combination of pultrusion technology with the chain of carbonization in one process for the production of unidirectionally reinforced C/C profiles. The CF rovings are infiltrated via an injection box so that emissions could be greatly reduced and a controlled impregnation could be realized. In addition, the deflection-free guidance of the CF during impregnation reduces possible distortion of the final profiles during carbonization due to unfavorably aligned CF. Therefore, carbonization is performed in an open mold. The use of the injection box enables the economic production of additional profile shapes. Furthermore, we included a production study on pultrusion, as well as mechanical and microstructural investigations of the resulting material.

## 2. Materials and Methods

### 2.1. Raw Materials

For the processing trials, materials are used as they are usually applied in the pultrusion process. The fiber material is a CF roving of the type PX35 (50K) with 3750 tex from the manufacturer ZOLTEK$^{TM}$ Europe in Nyergesujfalu, Hungary. Table 1 shows the components of the phenolic resin system based on a resol–novolak mixture and an internal mold release.

**Table 1.** Phenolic resin system.

| Component | Mixing Ratio | Component Name | Supplier |
|---|---|---|---|
| Phenolic resin | 100 | PF 7541 FW | Bakelite®, Iserlohn, Germany |
| Internal mold release | 3 | PAT®-659/A | E. und P. Würtz GmbH & Co. KG, Bingen am Rhein, Germany |

### 2.2. Manufacturing Process

Figure 2 shows the pultrusion setup. For the trials, a classic injection box was used. The main advantage here is the lower emission compared to an open resin bath. At the same time, cost-effective tools and classic system peripherals can be used in comparison to microwave curing. In addition, the process can be made cleaner, and it is possible to achieve more controlled impregnation with smaller quantities of material. The main task of the injection box is the homogeneous impregnation of the dry reinforcing fibers and preforming of the profile geometry. Due to the special geometric design, a steadily increasing impregnation pressure builds up to the mold transition. At the same time, care must be taken to ensure the thermal decoupling from the heated mold in order to prevent premature curing of the matrix system. For the production of the sample profiles, 32 CF rovings were used to fabricate a UD-reinforced profile. In agreement with the material manufacturers, the values for mold temperatures and haul-off speed listed in Table 2 were selected as initial process parameters. After a short startup period, a good profile quality with a smooth surface and cured, fully impregnated fibers could be produced after only a few meters. This quality was also reproducible over about 20 m before the surface quality deteriorated due to resin deposits. Optimization of the process parameters was not carried out in this trial.

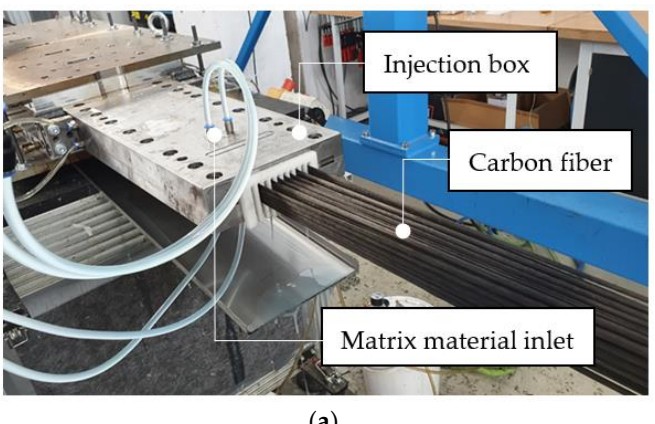

(**a**)

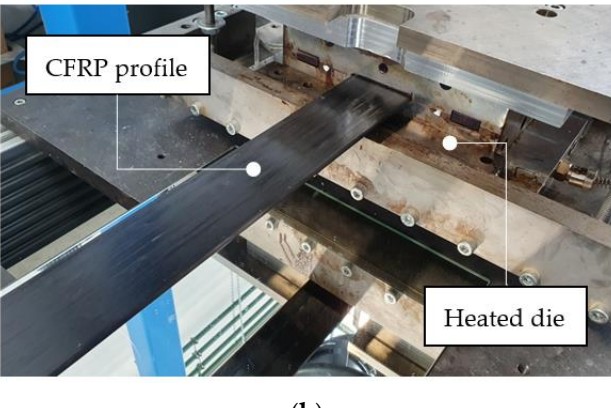

(**b**)

**Figure 2.** Pultrusion line setup: (**a**) CF inlet; (**b**) pultruded CFRP profile.

**Table 2.** Process parameters.

| Parameter | Value |
|---|---|
| Length of die [mm] | 1000 |
| Cross section [mm × mm] | 50 × 2 |
| Number of heating zones [-] | 5 |
| Process speed [mm/min] | 600 |
| Temperatures in heating zone 1–5 [°C] | 200-200-220-220-220 |

The pyrolysis of the CFRP composites to C/C composites was carried out in the pyrolysis, siliconization, and sintering furnace FCT-FS W 315/800-2400-PS (FCT Anlagenbau GmbH, Sonneberg, Germany) up to a final temperature of 1000 °C. The exact temperature-time profile for operating the furnace is shown in Figure 3. After initial evacuation of the furnace, the main process was realized mainly with argon purging. The cooling was achieved passively.

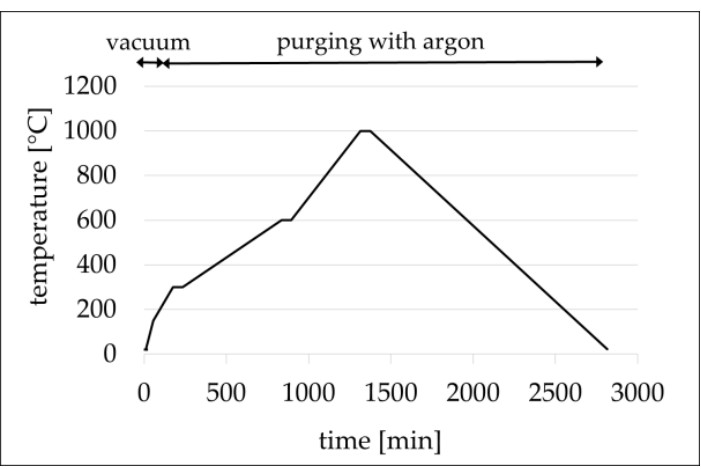

**Figure 3.** Temperature–time profile of the pyrolysis process.

*2.3. Characterization Methods*

The determination of the achieved fiber volume content $\varphi$ was calculated using the following correlations:

$$\varphi = \frac{V_F}{V_C} = \frac{N_F \; Tt}{A_C \; \rho_F} \tag{1}$$

where $V_F$ is the volume of the fibers, $V_C$ is the volume of the composite, $N_F$ is the number of rovings, $Tt$ is the linear mass density of fibers in g per 1000 m, $A_C$ is the cross-section area of the composite, and $\rho_F$ is the density of the fiber material. The material data provided by the manufacturers were used.

The roughness measurement on the pultrusion profiles was performed with the MarSurf Perthometer M2 (Mahr GmbH, Göttingen, Germany) on CFRP samples as well as on samples after pyrolysis. A glass-fiber-reinforced epoxy resin profile produced with the same pultrusion tool was used as reference for the roughness measurements. These measurements were realized for all materials in both the 0° and 90° fiber direction (five single measurements per direction and sample).

The changes in dimensions and in mass during the pyrolysis process were determined for 12 specimens (200 mm × 50 mm × 2 mm) using a caliper gauge and a balance.

The apparent porosity measurement and the determination of the bulk density of the CFRP and C/C composites were carried out in accordance with the DIN EN ISO 18754 (2022) standard by the method of liquid displacement. In each case, six specimens with the dimension 20 mm × 20 mm × 2 mm were used.

To characterize the microstructure, a longitudinal section and a cross section of the specimens were prepared by cold embedding in epoxy resin for examination by light microscopy (Axio Scope from Carl Zeiss AG, Oberkochen, Germany).

Four-point bending tests were performed following the DIN EN ISO 14125 (2011) standard to characterize the mechanical properties of the CFRP and C/C composites. In each case, eight specimens with the dimension of 100 mm × 15 mm × 2 mm were used. A Zwick/Roell Z5.0 standard testing machine (ZwickRoell GmbH & Co. KG, Ulm, Germany) with a load cell of 5 kN was used to conduct the four-point bending tests. The radius of the supports and the central loading members were 2 mm. The support width was 81 mm, and the inner span width was 27 mm. The test speed was 5 mm/min. The flexural modulus was determined as secant modulus within the strain ranges from 0.05 to 0.25% and from 0.05 to 0.13% for the CFRP and C/C composites, respectively.

The macro images of the pultruded and pyrolyzed specimens, as well as those of the fractured flexural specimens, were obtained using the OM-D E-M5 camera (OM Digital Solutions GmbH, Hamburg, Germany).

## 3. Results

During the production trials of the pultrusion process, good optical results were achieved in terms of plane and smooth surface quality and the degree of impregnation and cure. The production process could be implemented for a long period of time and with reproducible results. Defect patterns such as surface deposits could be minimized through typical purge steps in production. The trials were carried out with 32 CF rovings, resulting in a fiber volume content of 66.7%.

Figure 4 shows the pultruded products with UD reinforcement after pyrolysis. The products remained dimensionally stable without delamination or causing the structure to swell. After pyrolysis, the composites also remained dimensionally stable. Structural defects are not visible.

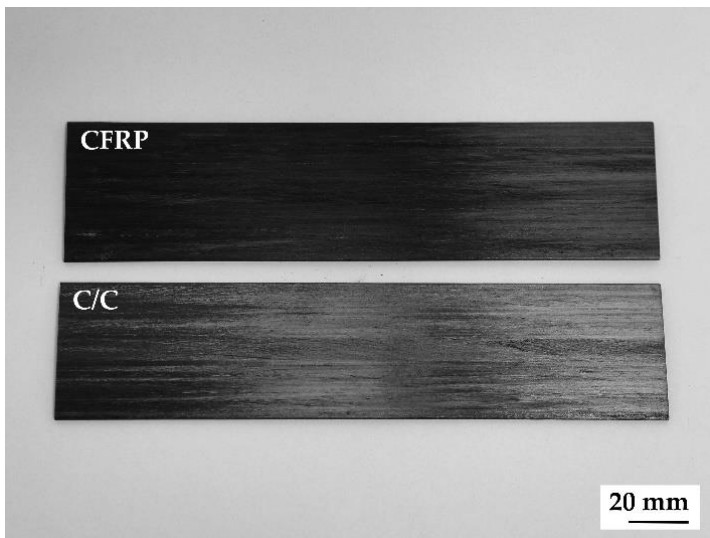

**Figure 4.** Macro images of the pultruded (CFRP composites) and pyrolyzed specimens (C/C composites).

Figure 5 presents the results of the roughness measurement along the pull-off direction (0°—longitudinal to the fiber direction) as well as vertical to it (90°—transversal to the fiber direction). In addition to the CFRP and C/C specimens, a reference value (glass fiber-reinforced epoxy from the same tool) is also shown for comparison purposes. The higher roughness in the CFRP composite is due to the use of phenolic resin, and the higher roughness in the C/C composite is due to the subsequent pyrolysis. Independent of the test direction, increases in roughness of approx. 56–65% were determined due to the pyrolysis step.

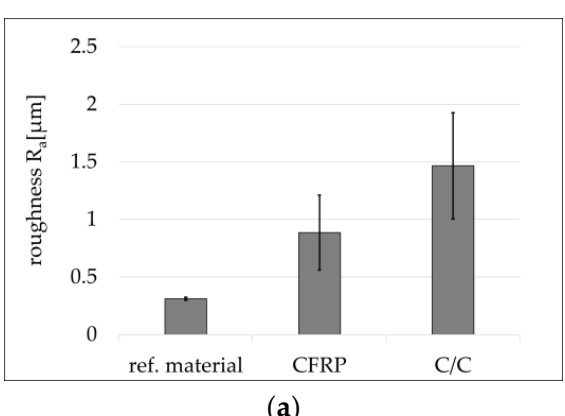
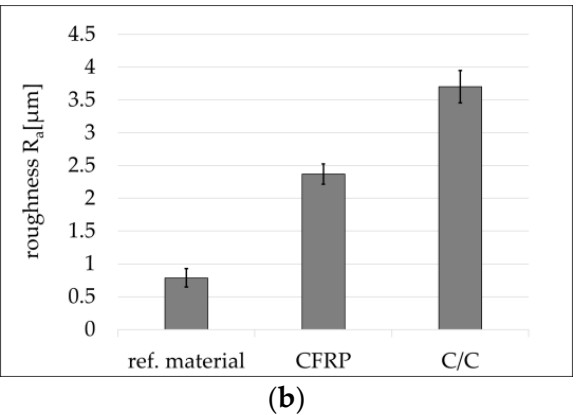

**Figure 5.** Roughness for different development stages: (**a**) longitudinal direction; (**b**) transversal direction. Reference material is glass-fiber-reinforced epoxy from the same tool.

The observation of the light microscopy images of the CFRP reveals a homogeneous microstructure containing both open and closed porosity. The porosity most likely originates from the condensation products of the crosslinking reaction during the polymer curing process (Figure 6).

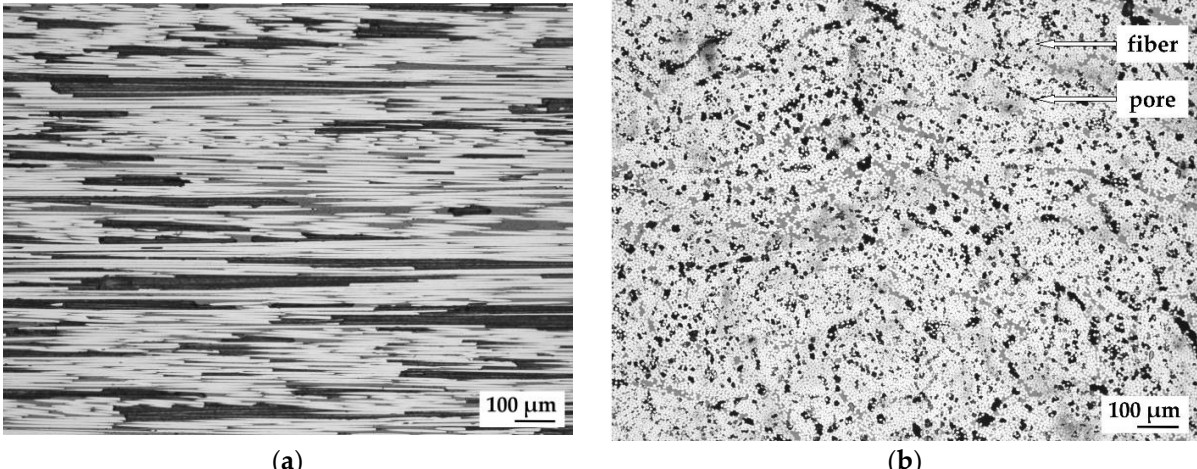

(**a**) (**b**)

**Figure 6.** Microstructure of the CFRP composite (light microscopy, bright field): (**a**) longitudinal section; (**b**) cross section (light color: fiber areas; dark color: pores).

The result of the microstructural analysis of the C/C specimen is rather similar to that from the previous CFRP stage. Moreover, the C/C has a more or less homogeneous distribution of C/C areas and pores with increased visible closed pores (Figure 7b) compared to CFRP but still without delamination. However, no regular crack pattern is formed because the UD composites shrink perpendicular to the direction of reinforcement, which counteracts the crack formation (Figure 7).

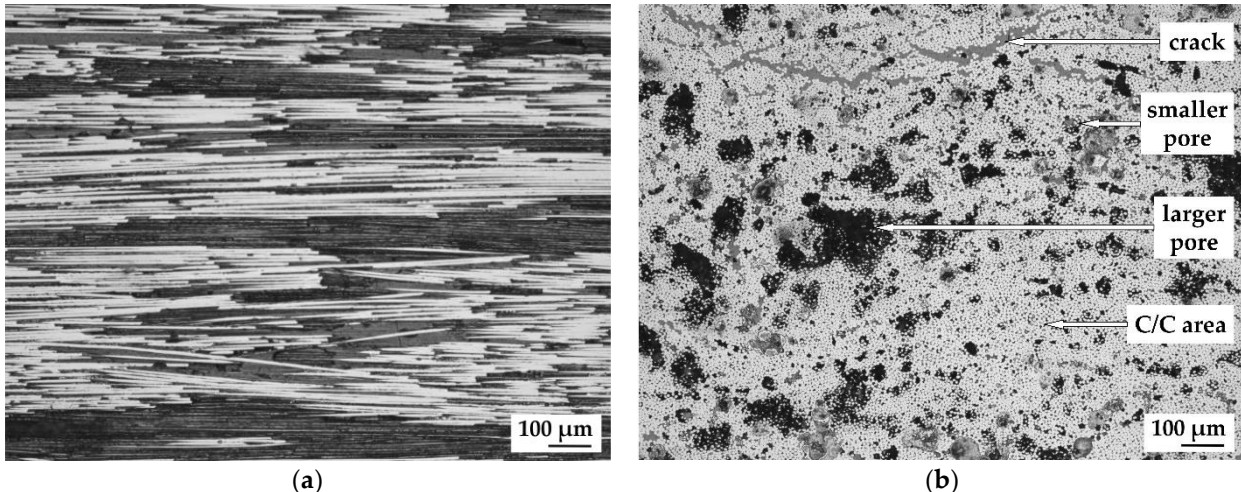

(**a**) (**b**)

**Figure 7.** Microstructure of the C/C composite (light microscopy, bright field): (**a**) longitudinal section; (**b**) cross section (light color: fiber areas; dark color: pores; gray: cracks filled with epoxy-embedding material).

Figure 8 shows the relative change in dimensions and mass during pyrolysis. Furthermore, the bulk density and the apparent porosity of the CFRP and the C/C samples are displayed. The pyrolysis of CFRP to C/C results in a mass reduction of approx. 10%. The shrinkage of the specimens takes place only in the thickness and width direction. There is no change in the length direction. The bulk density is slightly higher in the C/C composite

compared to the CFRP, whereas the apparent porosity in the C/C composite is lower than in CFRP.

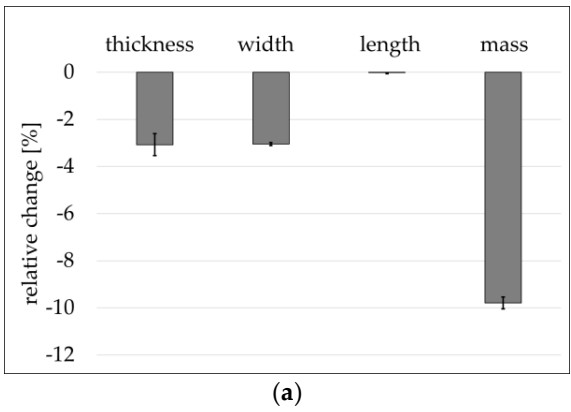 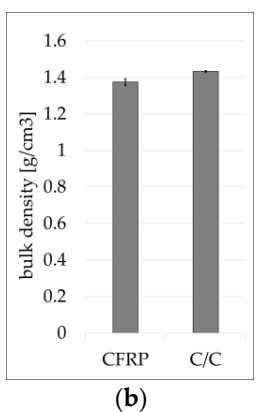 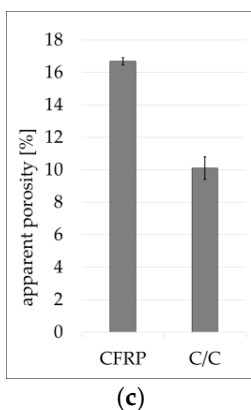

(a)        (b)        (c)

**Figure 8.** (**a**) Relative change in dimensions and in mass during pyrolysis. (**b**) Bulk density of the CFRP and C/C samples. (**c**) Apparent porosity (volume of open pores related to the bulk volume) of the CFRP and C/C samples.

Figure 9 shows the mechanical properties of the CFRP and the C/C composites determined by four-point bending tests with typical stress–strain curves. The pyrolytic conversion of CFRP to C/C results in a reduction of flexural strength and strain at a break of approx. 80% for both values; the flexural modulus remains at the same level within the standard deviation though.

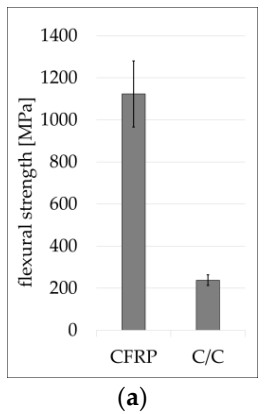 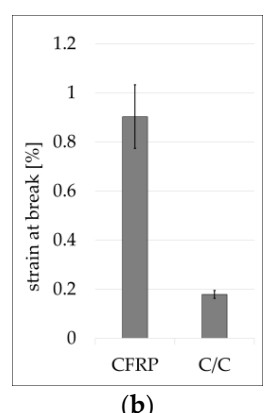 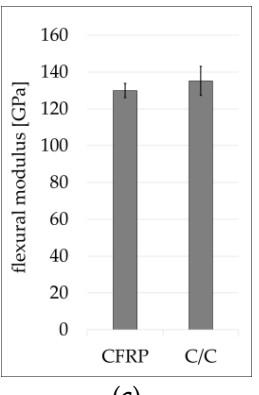 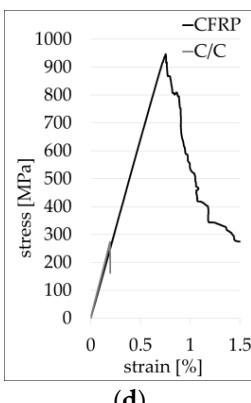

(a)       (b)       (c)       (d)

**Figure 9.** Mechanical properties of CFRP and C/C composites determined by four-point bending tests: (**a**) flexural strength; (**b**) strain at break; (**c**) flexural modulus; (**d**) typical stress–strain curves.

The CFRP and C/C composites show linear elastic behavior up to the point of failure. No premature matrix fractures are visible in the curves. The fibers have a higher stiffness and strength compared to the matrix. The main load is carried by the fibers which are aligned in load direction. The macroscopic behavior of the composites is fiber-dominated. The CFRP composites show a successive failure after reaching the maximum stress. The C/C composite shows higher brittleness but can still take residual load after failure.

The failure of the CFRP composites occurs mainly due to compressive stresses and interlaminar shear stresses, whereas the C/C composites fail due to tensile stresses (Figure 10).

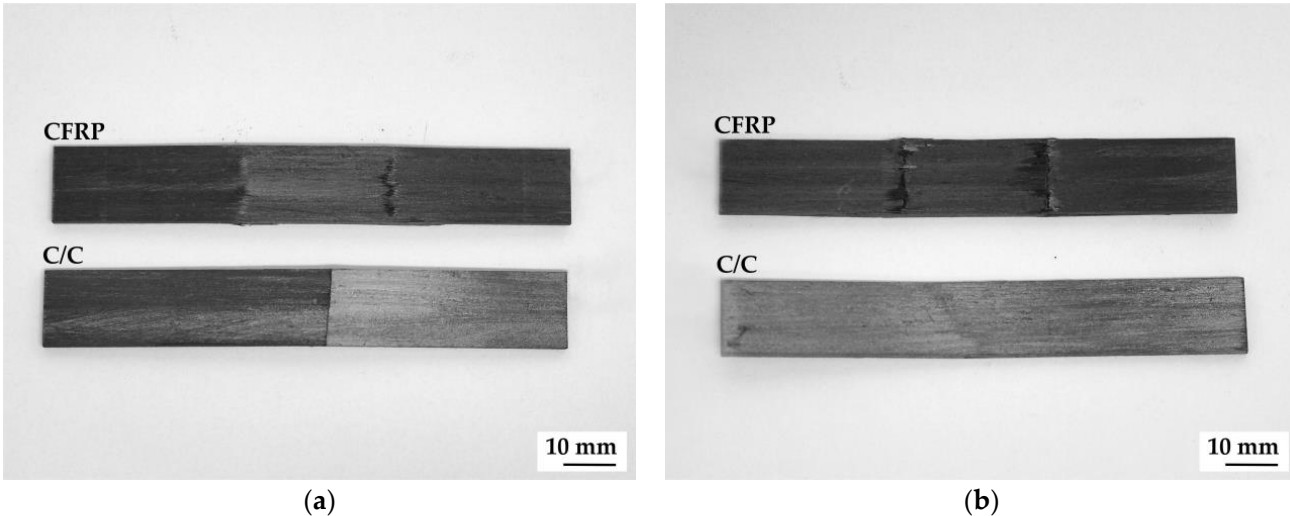

**Figure 10.** Macro images of fractured flexural specimens of the CFRP and C/C composites: (**a**) tensile-loaded side; (**b**) compression-loaded side.

## 4. Discussion

The overall pultrusion process of the production of a CFRP specimen suitable for the C/C production is well functioning. The selected pultrusion parameters (Table 2) resulted in a successful crosslinking without swelling and non-impregnated areas, and it also resulted in visually appealing, dimensionally stable components. The polycondensation reaction of the crosslinking leads to the formation of microporosity (Figure 6) because the water as a condensation product cannot evaporate. There are several approaches to reducing porosity. On the one hand, vacuum extraction could be used in the molding tool, and on the other hand, the pressure during impregnation could be increased. Both changes to the system are currently being examined. However, an increase in pressure in particular is not a trivial parameter adjustment and may not be feasible without major rebuilds.

According to Schulte-Fischedick et al. [22], pyrolysis (of CF-reinforced phenolic resins) can be divided into four stages. First (1) is the crosslinking stage (up to approx. 300 °C), in which further crosslinking with simultaneous separation of monomers and release of solvents takes place. Second (2) is main pyrolysis (300–500 °C), by which the matrix polymer is converted into carbon. In this step, a strong loss of mass and a high volume shrinkage occurs, resulting in a loose network of conjugated carbon domains, which are connected by hydrogen bonds. Third (3) is dehydrogenation (500–1200 °C), by which the hydrogen present in the structure is removed. The linear conjugated carbon domains are linked and form a coherent carbon structure. Fourth (4) is defect healing (above 1200 °C); most chemical processes are terminated at this stage and only individual structural defects are still being healed.

In this study, pyrolysis was conducted up to 1000 °C, which means that the dehydrogenation was not fully conducted, but the majority of the conversion has already been completed. Due to additional economic reasons with regard to the targeted LSI process [23], the maximum temperature of the pyrolysis was limited. After carbonization, however, most of the fission products have been split off, which leads to the formation and enlargement of the shown pores and cracks, as well as the macroscopic shrinkage in width and thickness. The conversion of the polymer matrix to carbon results in an increase in density and large shrinkage. Therefore, the dimensions of the sample are reduced by that shrinkage. This is also the explanation as to why the apparent porosity (ratio of the total volume of the open pores to its bulk volume) decreases, because the UD-oriented fiber structure does not prevent shrinkage in the thickness and width direction.

The dimensional changes after carbonization, as well as the increase of porosity, align with those presented in Ref. [21]. There, the distortion during carbonization was

significantly reduced with the use of a closed graphite mold. In the case of the flat specimens produced in this study, no distortion was detected during carbonization in an open crucible. In addition, the shrinkage in the former study is higher compared to that in our study, and the porosity is lower. In Ref. [21], no statements are made about the density, but due to the shrinkage behavior and the porosity changes, a similar density change is assumed.

Further optimization should be performed on the material side because in preliminary tests of the pultrusion process, setup and optimization were performed without considering the subsequent process of pyrolysis to C/C. There are several resin properties that are important for the pultrusion process, such as flowability, wettability of the fiber reinforcement, crosslinking and reaction rate at different temperatures, and the low proportion of volatile products. Furthermore, for the pyrolysis process, different properties—such as dimensional stability, high carbon yield, low volume shrinkage, low amount of volatile decomposition products, porosity formation, and, in particular, formation of open porosity for possible post-infiltration steps—are of importance. A resin material that fits to both processing steps is probably not commercially available.

A possible optimization can also be seen in the multidirectional fiber arrangement during the pultrusion process. In addition to the use of UD rovings, random fiber mats, textile semi-finished products such as woven fabrics with two fiber orientations, or even non-crimp fabrics with additional fiber angles would also be conceivable. Using those could change the shrinkage behavior in the respective dimensions during pyrolysis, which is compensated by the formation of a regular crack system accessible from the outside. This also changes the outgassing behavior during pyrolysis and the formation of closed pores. However, when textile semi-finished products are used, both the resin and the infiltration process in the pultrusion process must be adapted to the textile in question. It must be borne in mind that, unlike rovings, a textile semi-finished product is already relatively compact when it enters the injection box. This requires, for example, a reduction in the viscosity of the resin or even an increase in pressure in the injection box. In the case of non-crimp fabrics, the binder filament is removed during pyrolysis, thus forming porosity channels in the C/C, which can serve as a flow channel for a subsequent infiltration with silicon. In this way, silicon carbide reinforcement in the thickness direction (z-direction) can also be realized. However, the material of the binder yarn should have a higher melting point than the pultrusion tool, so that the fiber architecture remains intact.

## 5. Conclusions

In this study, a large-scale process chain starting with the raw material selection, follow by the pultrusion and the final pyrolysis was successfully implemented. The CFRP composites with UD CF reinforcement in a phenolic matrix and very high fiber volume content were produced. These composites remain dimensionally stable during the pyrolysis process. Further optimization of the process, the material, and the fiber arrangement should significantly improve the composites in terms of porosity and flexural strength. This technology thus enables the cost-effective production of ceramic matrix composites. Based on the gained knowledge, it should also be possible to implement more complex profiles such as angles, U-profiles, or even hollow profiles such as tubes with braided reinforcement. In the next step, the use of multiaxial fiber reinforcements for a more homogeneous crack structure in the C/C material should also be assessed. This externally accessible porosity is then to be infiltrated with molten silicon to generate a silicon carbide matrix. It would then be possible to produce long, continuously reinforced CMC profiles economically.

**Author Contributions:** Conceptualization, K.R. and D.L.; methodology, J.H.M.S., D.L. and K.R.; validation, K.R., D.L. and J.H.M.S.; investigation, K.R., J.H.M.S., M.K. and D.L.; writing—original draft preparation, J.H.M.S., K.R., D.L. and M.K.; writing—review and editing, L.K., D.N. and W.-G.D.; supervision, L.K., D.N. and W.-G.D. All authors have read and agreed to the published version of the manuscript.

**Funding:** This research received no external funding.

**Data Availability Statement:** Not applicable.

**Acknowledgments:** The authors want to thank the entire lab staff for their support regarding the technical implementation of the processes as well as for their support in the analyses.

**Conflicts of Interest:** The authors declare no conflict of interest.

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
