# Peer review of "Combining Pultrusion with Carbonization: Process Analysis and Material Properties of CFRP and C/C"

_ceramics, doi:10.3390/ceramics6010020_

Round 1

Reviewer 1 Report

The paper introduced one strategy  to make carbon fiber reinforced carbon composites (C/C) from polymer matrix composites (PMC) fabricated from pultrusion method, and compared the structure and properties of the resultant C/C and PMC, which is interesting. However, pultrusion is one process commonly used to make PMC, and heat treatment of polymer in inert atomsphere to get carbon matrix is also commonsense.

1.     The importance and the novelty of this trial should be emphasized clearly.

2.     The behaviour of phenolic resin during the heat-treatment should be studied to explain some results of C/C composite.

Reviewer 2 Report

Comments for manuscript ID ceramics-2100343

The following comments are recommended to improve the manuscript:

1.    The title should be modified.

2.    Supporting the abstract with more data and descriptive sentences is recommended

3.    The novelty should be clearly described.

4.    No comparison between obtained data and literature was found in the text.

5.    The abbreviation should be corrected.

6.    Have the decrease of bulk density and apparent porosity been similarly reported in the literature?

7.    The Figures should be developed.

8.    What is “s.” in “s. Figure”?

9.    The reference list is not uniform.

10. English improvement is recommended.

Round 2

Reviewer 2 Report

Comments for manuscript ID ceramics-2100343-v2

1.    No need to bring “see” in “see Figure”.

2.    The abbreviations still need correction: e.g. UD, PEEK, RTM and …

Please carefully see all the abbreviations in your text.

3.    The English style of your text needs improvements yet. As examples some points are mentioned as follows:

·       The sentence in line 21 “The resulting …”

·       The sentence in line 96 “Due to their exceptional fire….”

·       The sentence in line 199 “Figure 4  shows …”

Please carefully go through your text and improve the English language.

4.    The figures should be more developed. As examples some points are mentioned as follows:

·       In figure Fig. 8a: the labels can be transferred to the top.

·       All charts should have borders.

5.    The references are not uniform yet.

Author Response

please view attached document
